

# CliffDelineaTool v1.1.0: an algorithm for identifying coastal cliff base and top positions

Zuzanna M. Swirad, Adam P. Young

Scripps Institution of Oceanography, University of California San Diego, 9500 Gilman Dr., La Jolla, CA 92093, USA

*Correspondence to:* Zuzanna M. Swirad (zswirad@ucsd.edu)

**Abstract**. Correct quantification of coastal cliff erosion requires accurate delineation of the cliff face bounded by the cliff top and base lines. Manual mapping is time consuming and relies on mapper's decisions and skills. Existing algorithms are generally site specific and may be less suitable for areas with diverse cross-shore cliff geometry. Here we describe *CliffDelineaTool* (v1.1.0), a MATLAB-based algorithm that identifies cliff base and top positions on

complex cliffs using cross-shore transects extracted from digital elevation models. Testing on four 750-1200 m cliffed coastlines shows that the model performance is comparable to manual mapping and provides some advantages over existing models but provides poor results for cliff sections with ambiguous cliff top edges. The results can form the basis for a range of analyses including coastal inventories, erosion measurements, spatio-temporal erosion trends, and coastline evolution modelling.

**1 Introduction**

Correct quantification of cliff erosion for scientific and management purposes requires accurate delineation of coastal cliff faces. Cliff base and top positions are often digitized manually on georeferenced maps, aerial photographs, ortho-photographs, and digital elevation models (DEMs) (e.g. Dornbusch et al., 2008; Hapke et al., 2009; Brooks et al., 2012; Orviku et al., 2013; Swirad et al., 2017; Young, 2018). However, manual mapping is subject to the mapper's

decisions and skills (Moore, 2000). The lack of uniform definitions for a cliff base and top leads to further inconsistencies. Payo et al. (2018) suggested that for consistency, the mapping should be performed at the same time for the entire dataset, and by one mapper. However, this becomes problematic for multi-temporal studies, and those that build on previous efforts.

While manual digitization may be necessary when using cartographic sources, DEMs provide an opportunity to map

the cliff base and top programmatically as these features are characterized by local changes in slope (Liu et al., 2009). Preference of manual over automated mapping, particularly for small (<km) areas of interest, may result from clear visual recognition of cliff base and top positions, and challenges with developing an algorithm that works for a range of cliff geometries outside the initial calibration dataset. However, automated cliff delineation increases objectivity and consistency, and decreases processing time, which is particularly useful for large-area high-resolution topographic

datasets (Swirad and Young, 2021).

Several studies used automated or semi-automated techniques to separate the cliff face from foreshore and/or hinterland. The studies vary in terms of local settings, spatial scale, available source datasets, and purpose. For instance, Alessio and Keller (2020) extracted the cliff base as a 3 m (NAVD88) elevation contour. Richter et al. (2013)



applied terrain filters (1st and 2nd derivative of elevation) sensitive to slope change to identify the cliff base line for a

simple cliff morphology. Several authors identified cliff base positions as inflection points along cross-shore transects. Liu et al. (2009) identified cliff base and top points using transects combined with image segmentation, surface reconstruction, and edge detection on ortho-images. For relatively simple cliffs, Terefenko et al. (2019) identified the cliff base as the seaward-most location along the transects with at least 0.5 m vertical change over 1 m horizontal distance. Palaseanu-Lovejoy et al. (2016) extracted cliff base and top positions along cross-shore transects by

comparing transect elevations with elevations along a straight line between transect ends ('trendline'). Cliff base and top points were defined as locations along the transects with the largest vertical distance between the cliff profile and trendline, with the cliff base located below and cliff top above the trendline. Transect lengths were manually adjusted to ensure proper cliff top selection (Palaseanu-Lovejoy et al., 2016). Payo et al. (2018) developed the *CliffMetrics* algorithm using the model of Palaseanu-Lovejoy et al. (2016) combined with automated extraction of the shoreline

and transects for coasts with complex alongshore geometry where small bays and headlands alternate. They used a constant transect length with decrease in model performance, but considerable time gain (Payo et al., 2018). *CliffMetrics* performs well for simple cross-shore cliff morphology, but it is less suitable for more complex cliff profiles, where rotational landslides, within-cliff flattening, roads, etc. are present (Swirad and Young, 2021).

Here, we build on previous models to develop a new MATLAB-based algorithm, *CliffDelineaTool* (v1.1.0; Swirad,

2021) that identifies cliff base and top positions on cross-shore transects for a range of complex cliff geometries. The model parameters are calibrated using four cliff sections that encompass a range of geomorphic settings, and then tested on four different cliff sections with topography ranging from simple to complex. The results are compared to manually mapped cliff lines and *CliffMetrics*.

## 2 Methods

### 2.1 *CliffDelineaTool* workflow

The model (Figure 1) uses eight user-defined parameters (Table 1) and an input text file containing rows of point ID, transect ID, distance from the seaward end of the transect, and elevation (Swirad et al., 2016). A single input file includes multiple ordered transects representing an entire coastal section.





**Table 1.** *CliffDelineaTool* **user-defined parameters, and values optimized for the calibration coastal sections (areas of interest (AOIs) #1-4) and used to validate the model (AOIs #5-8).**

| Parameter | Description | Calibration | | | | Validation | | | |
|---|---|---|---|---|---|---|---|---|---|
| | | AOI #1 | AOI #2 | AOI #3 | AOI #4 | AOI #5 | AOI #6 | AOI #7 | AOI #8 |
| *MaxBaseElev* | Maximum elevation of the cliff base (m, NAVD88) | 5 | 9 | 5 | 5 | 5 | 5 | 7 | 5 |
| *NVert* | Number of consecutive points to define local scale | 20 | 20 | 30 | 30 | 20 | 20 | 20 | 20 |
| *BaseSea* | Maximum seaward slope angle at the cliff base (°) | 12 | 13 | 9 | 15 | 20 | 20 | 20 | 15 |
| *BaseLand* | Minimum landward slope angle at the cliff base (°) | 34 | 4 | 6 | 29 | 30 | 20 | 30 | 20 |
| *TopSea* | Minimum seaward slope angle at the cliff top (°) | 39 | 32 | 18 | 14 | 30 | 25 | 30 | 40 |
| *TopLand* | Maximum landward slope angle at the cliff top (°) | 14 | 8 | 10 | 19 | 10 | 25 | 20 | 35 |
| *PropConvex* | Threshold parameter for stage 2 cliff top replacement | n/a | 0.4 | 0.1 | 0.6 | 0.2 | 0.8 | 0.5 | 0.5 |
| *SmoothWindow* | Moving window size for cross-shore location smoothing (number of transects) | 8 | 4 | 13 | 14 | 10 | 5 | 5 | 25 |



**SET UP THE ALGORITHM**

Load the file

Set parameter values (Table 1)

Identify columns of cross-shore distance and elevation

**PREPARE TRANSECTS**

Fill elevation data gaps

Calculate local slope angles

Limit transects landwards to the highest point + *NVert*

Create trendline #1 (line connecting transect ends)

Calculate vertical distance between cliff profile and trendline #1

**IDENTIFY CLIFF BASE**

Identify potential cliff base locations (elevation - trendline #1 < 0, > MaxBaseElev, < BaseSea, > BaseLand)

Select cliff base location (potential cliff base with max vertical distance between trendline #1 and elevation)

**Cliff Top Stage 1: INITIAL CLIFF TOP**

Create trendline #2 (line connecting cliff base and landward transect end)

Identify potential cliff top locations (elevation - trendline #2 > 0, > *TopSea*, < *TopLand*)

Select cliff top location (potential cliff top with max vertical distance between trendline #2 and elevation)

**Cliff Top Stage 2: SHIFT CLIFF TOP LANDWARD?**

*Are there alternative potential cliff top locations > NVert landwards from the stage 1 cliff top?*

YES
*Vertical distance between the elevation and trendline #2 > PropConvex × that of the stage 1 cliff top?*

NO
Retain stage 1 cliff top

YES
*Point located above trendline #3?*

NO
Retain stage 1 cliff top

YES
Select cliff top with largest vertical distance from trendline #2

NO
Retain stage 1 cliff top

**Cliff Top Stage 3: REMOVE ALONGSHORE CLIFF TOP OUTLIERS**

Define smoothed cliff top using median alongshore moving window (*SmoothWindow*)

Find cross-shore difference between cliff top and smoothed cliff top

Flag transects with outliers (standardized residual > 2)

Find potential cliff top closest to the smoothed cliff top location

*Is its standardized residual > 2?*

YES
Remove cliff top location; transect has no modelled cliff top ('skipped transect')

NO
Replace modelled cliff top

**Figure 1.** *CliffDelineaTool* **workflow and the three cliff top processing stages. Calibrated parameters** *NVert, MaxBaseElev, BaseSea, BaseLand, TopSea, TopLand, PropConvex, SmoothWindow* **are defined in Table 1.**



Processing is performed on a transect-by-transect basis. Transect elevation gaps are filled through extrapolation
(transect peripheries) and linear interpolation (interior sections). For each point, local seaward and landward slope
angles are calculated as an average slope between the point and a user-defined number of adjacent points called the
*NVert* parameter. The *NVert* value is used at various stages of the model workflow to determine local spatial
relationships between points (Swirad and Rees, 2015). To remove unnecessary inland points, the landward transect
end is set to *NVert* points landward of the highest elevation. Next, a straight line ('trendline #1') is created by
connecting transect ends (Figure 2a; after Payo et al., 2018).

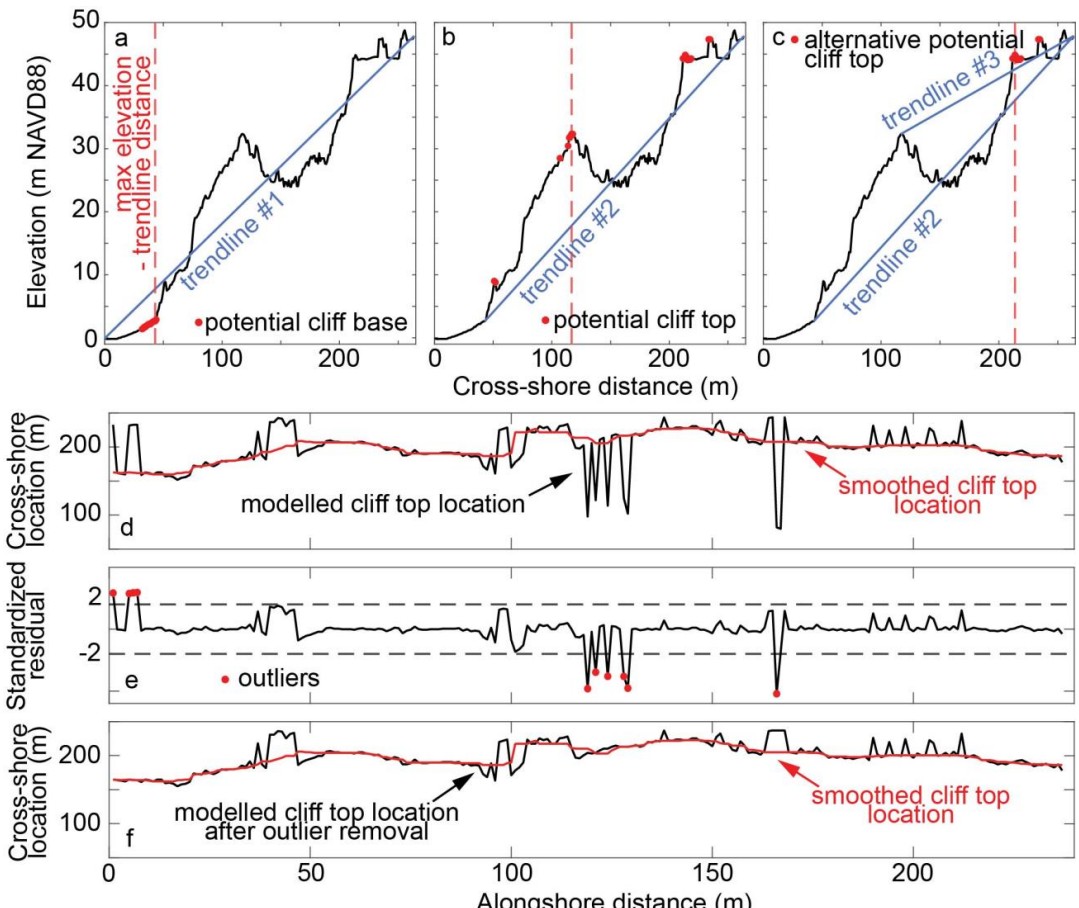

**Figure 2. Example identification of cliff base and top positions (transect #123 of AOI #3): a) potential cliff base positions
and the point with the largest distance from trendline #1; b) potential cliff top positions and the point with the largest
distance from trendline #2 (stage 1); c) alternative potential cliff top positions in case of complex cross-shore profile and the
point with the largest distance from trendline #2 (stage 2). Identification and removal of alongshore cliff top outliers (stage
3) – alongshore distribution of d) stage 2 modelled and smoothed cliff top; e) standardized residuals between smoothed and
stage 2 modelled cliff top; f) stage 3 modelled cliff top with outliers removed.**

Potential cliff base locations are below trendline #1 and fulfill user-defined criteria of maximum elevation of the cliff
base (*MaxBaseElev*), maximum local seaward slope (*BaseSea*) and minimum local landward slope (*BaseLand*). These
criteria eliminate physically invalid cliff base elevations and ensure finding inflection points (concavities) at the full



transect (relationship to the trendline) and local (seaward and landward slopes) levels. From the set of potential cliff base locations, the point with the largest vertical distance from the trendline is selected to represent the cliff base following Palaseanu-Lovejoy et al. (2016) and Payo et al. (2018) (Figure 2a). If no points fulfill the criteria, the transect is skipped at this stage.

Cliff top identification consists of three stages (Figure 1). In stage 1, trendline #2 is created by connecting the modelled cliff base with the landward end of the transect (Figure 2b). Potential cliff top locations are located above trendline #2 and fulfill user-defined criteria of minimum local seaward slope (*TopSea*) and maximum local landward slope (*TopLand*) defining local convexities. From the set of potential cliff top locations, the point with the largest vertical distance from trendline #2 is selected to represent the cliff top following Palaseanu-Lovejoy et al. (2016) and Payo et al. (2018) (Figure 2b). If no points along the transect fulfill the criteria, the transect is skipped and has no explicitly modelled cliff top.

In some locations with complex cliff face profiles, such as rotational landslides or mid-cliff roads, a local flattening within the cliff face may exist and cause an incorrect cliff top selection. To account for these situations, in stage 2 the model checks if any alternative potential cliff top exists landwards of the initial cliff top location identified in stage 1. Alternative potential cliff top positions located closer than *NVert* points from the initial top are rejected to ignore points that are likely part of the same convex section. Alternative positions must also be located above trendline #3 (a line from the initial cliff top to the landward transect end, Figure 2c), and greater than *PropConvex* (values between 0 and 1) multiplied by the elevation difference between the initial cliff top and trendline #2. For example, if *PropConvex* = 0.5 and the initial cliff elevation is 4 m above trendline #2, potential cliff top locations must be >2 m (0.5 × 4 m) above trendline #2. If alternative positions exist, the initial top is re-located to the point with the largest vertical distance from trendline #2 (Figure 2c). Otherwise, the initial cliff top is retained.

In stage 3, cliff top outliers are identified by comparing the cross-shore location to an alongshore smoothed cliff line, created from the median of an alongshore moving window (*SmoothWindow*) (Figure 2d). Residuals are calculated as a distance between the smoothed and modelled cross-shore locations. Transects with standardized residuals (residual divided by residual standard deviation) >2 are flagged as outliers (Figure 2e) and re-examined. If the standardized residual of the potential cliff top closest to the smoothed cross-shore location <2, it replaces the previously modelled top. Otherwise, the transect has no explicitly modelled cliff top (Figure 2f).

## 2.2 Model development

### 2.2.1 *CliffDelineaTool* calibration

To optimize user-defined parameters (Table 1), four calibration coastal cliff sections in California (areas of interest (AOIs) #1-4) were selected spanning 650-1200 m alongshore that encompass a range of geomorphic settings (Table 2; Figure 3a-d). Topographic information for each AOI was derived from 1 m DEMs created from a 2016 airborne LiDAR dataset (Swirad and Young, 2021).





**Table 2. Characteristics of the areas of interest (AOIs) used to calibrate *CliffDelineaTool*. Average values are represented by mean ± standard deviation.**

|  | AOI #1 | AOI #2 | AOI #3 | AOI #4 |
|---|---|---|---|---|
| Geographic location | 34°28'15.54''N 120°13'8.30''W | 33°18'49.31''N 117°29'7.61''W | 33°20'47.17''N 117°31'20.70''W | 35°57'24.90''N 121°28'57.90''W |
| Alongshore extent (m) | 700 | 965 | 1180 | 655 |
| Number of transects | 141 | 194 | 237 | 132 |
| Cross-shore extent (m) | 87 | 159 | 264 | 249 |
| Average cross-shore cliff extent (m) | 23 ± 6 | 29 ± 15 | 153 ± 22 | 76 ± 32 |
| Average cliff base elevation (m, NAVD88) | 2.7 ± 0.4 | 6.7 ± 0.7 | 3.5 ± 0.7 | 4.3 ± 0.8 |
| Average cliff height (m) | 26 ± 2 | 23 ± 1 | 42 ± 2 | 60 ± 10 |
| Average foreshore slope (°) | 5.4 ± 5.4 | 5.7 ± 5.7 | 5.6 ± 3.6 | 7.1 ± 8.7 |
| Average cliff slope (°) | 47 ± 12 | 43 ± 18 | 28 ± 15 | 46 ± 13 |
| Average hinterland slope (°) | 10 ± 11 | 7.1 ± 10 | 16 ± 15 | 26 ± 17 |
| Morphology | Cliff is the unique slope, beach and hinterland are planar, cliff top has a near-straight shape | Back beach berm, gullies that intersect the cliff top, complex cliff top line | Rotational landslides resulting in within-cliff flattening, vegetation, beach cusps, road within cliff face | Rocky foreshore, road adjacent to the cliff, sloping hinterland, gullies |



**Figure 3. Areas of Interest (AOIs) used to calibrate (a-d) and validate (e-h)** *CliffDelineaTool*.

Calibration transects were spaced 5 m alongshore to capture meso-scale details of alongshore cliff geometry and sampled at 1 m cross-shore resolution. Points representing 'true' cliff base and top locations were visually selected





for each transect. Maximum cliff base elevation (*MaxBaseElev)* was subjectively set to 5 m (NAVD88) for AOIs #1, 3, and 4, and 9 m (NAVD88) for AOI #2 based on DEM inspection (Figure 3, Table 2). The remaining seven

parameters were calibrated to minimize root mean squared error (RMS) between true and modelled cliff base and top positions while not skipping too many transects. *NVert* and threshold slope angles (*BaseSea*, *BaseLand*, *TopSea* and *TopLand*) were calibrated using the characteristics of the true cliff base and top locations. Slope angles were calculated for a range of *NVert* values (Table 2). Local slope angle distributions were summarized in statistical terms (Figure 4), and outliers were defined as points greater than $q_3 + 1.5 \times (q_3 - q_1)$ or less than $q_1 - 1.5 \times (q_3 - q_1)$, where $q_1$ and $q_3$ are

the 25[th] and 75[th] percentiles (red pluses in Figure 4). *NVert* = 20 was selected for AOIs #1-2 and *NVert* = 30 was selected for AOIs #3-4 to minimize the number of outliers while maintaining a relatively narrow and normal slope angle distribution. Threshold slope angles were picked as minimum or maximum values excluding outliers (black whisker ends in Figure 4; Table 1).



**Figure 4. Distributions of the local seaward and landward slopes of true cliff base and top locations for varying *NVert* values. Boxplots include median (red bar), 25th ($q_1$) and 75th ($q_3$) percentiles (blue box), range excluding outliers (black whiskers) and outliers (red pluses). Outliers are defined as values greater than $q_3 + 1.5 \times (q_3 - q_1)$ or less than $q_1 - 1.5 \times (q_3 - q_1)$. Values in red located above the plots provide the number of outliers (if >0). Grey shadow indicates local slope distributions at selected *NVert* value.**

Next, *CliffDelineaTool* checks for false cliff tops caused by a local flattening within the cliff face using *PropConvex* (stage 2 of the model, Figure 1). *PropConvex* was calibrated by inspecting the cliff top RMS for values ranging from 0.1 to 0.9 at 0.05 intervals (Figure 5a-d). For the simple cliff morphology of AOI #1, introducing *PropConvex* did not change the modelled cliff top locations, but it did decrease the RMS for more complex AOIs #2-4 (Figure 5a-d; Table 1). Next, cliff top outliers were identified. Optimal *SmoothWindow* was selected by comparing moving window ranging from 1 to 20 alongshore transects for each AOI to minimize RMS and number of skipped transects (Figure



5e-h; Table 1). The modelled cliff base and top locations were converted to polylines using ArcGIS and intersected with transects to define cliff base and top locations on skipped transects. Overall, these steps improved automated cliff mapping performance for the more complex cliff sections and usually exhibited lower RMS compared to *CliffMetrics* (Table 3; Figure 6).

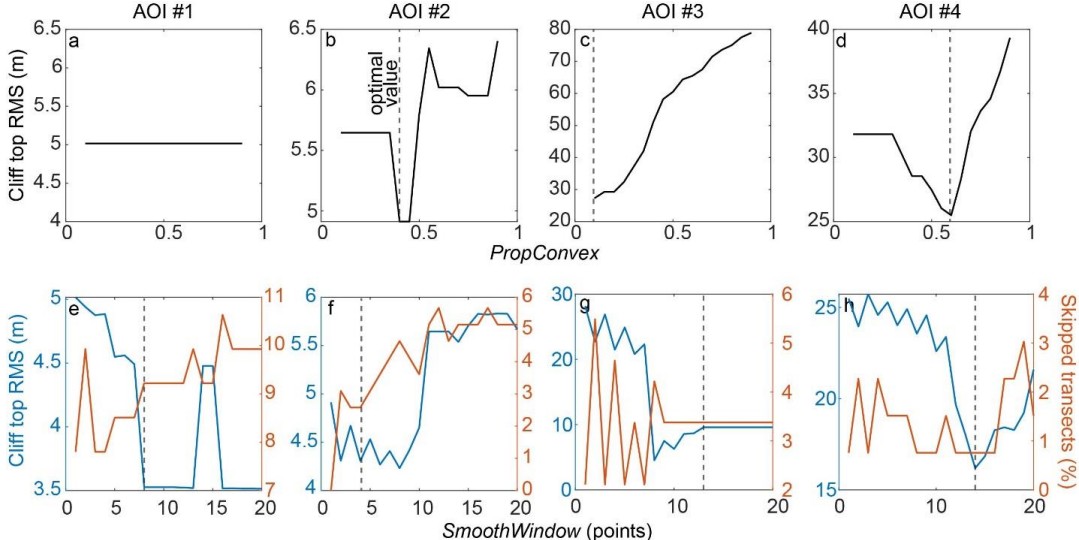

**Figure 5. Calibration of *PropConvex* (testing from 0.1 to 0.9 at 0.05 interval) (a-d) and *SmoothWindow* (testing from 1 to 20 transects) (e-h). Vertical dashed lines represent the optimal values (Table 1).**

**Table 3. Performance of *CliffDelineaTool* at various stages and *CliffMetrics* (Palaseanu-Lovejoy et al., 2016; Payo et al., 2018).**

| AOI | Error metric | CliffDelineaTool | | | | CliffMetrics |
|---|---|---|---|---|---|---|
| | | Stage 1 | Stage 2 | Stage 3 | Stage 3 including skipped transects | |
| #1 | Skipped cliff base points (%) | 0 | 0 | 0 | 0 | 0 |
| | Cliff base RMS (m) | 0.3 | 0.3 | 0.3 | 0.3 | 0.4 |
| | Skipped cliff top points (%) | 7.8 | 7.8 | 9.2 | 0 | 0 |
| | Cliff top RMS (m) | 5.0 | 5.0 | 3.6 | 6.1 | 5.0 |
| #2 | Skipped cliff base points (%) | 0 | 0 | 0 | 0 | 0 |
| | Cliff base RMS (m) | 5.9 | 5.9 | 5.9 | 5.9 | 14 |
| | Skipped cliff top points (%) | 0 | 0 | 2.6 | 0 | 0 |
| | Cliff top RMS (m) | 6.7 | 4.9 | 4.4 | 4.9 | 5.8 |
| #3 | Skipped cliff base points (%) | 2.1 | 2.1 | 2.1 | 0 | 0 |
| | Cliff base RMS (m) | 5.4 | 5.4 | 5.4 | 5.3 | 74 |
| | Skipped cliff top points (%) | 2.1 | 2.1 | 2.1 | 0 | 0 |
| | Cliff top RMS (m) | 83 | 28 | 9.7 | 10 | 74 |
| #4 | Skipped cliff base points (%) | 0.8 | 0.8 | 0.8 | 0 | 0 |
| | Cliff base RMS (m) | 1.7 | 1.7 | 1.7 | 2.0 | 31 |
| | Skipped cliff top points (%) | 0.8 | 0.8 | 0.8 | 0 | 0 |
| | Cliff top RMS (m) | 40 | 26 | 16 | 16 | 29 |



Figure 6. Location of the cliff base and top at various stages of the *CliffDelineaTool* calibration.

### 2.2.3 *CliffDelineaTool* evaluation

Ten geoscientists manually digitized the cliff base and top for four different sections of the California coastline (AOIs #5-8) with diverse morphology using 1 m resolution DEMs and hillshade maps (Table 4; Figure 3e-h). The 'true' cliff base and top positions were defined as the median of the manually mapped positions.





Table 4. Characteristics of the areas of interest (AOIs) used to evaluate *CliffDelineaTool*. Average values are represented by mean ± standard deviation.

| | AOI #5 | AOI #6 | AOI #7 | AOI #8 |
|---|---|---|---|---|
| Geographic location | 34°25'6.44''N 119°47'49.20''W | 33°21'19.13''N 117°32'3.34''W | 34°33'36.12''N 120°37'49.53''W | 34°54'6.71''N 120°39'36.70''W |
| Alongshore extent (m) | 845 | 1195 | 745 | 1075 |
| Number of transects | 170 | 240 | 150 | 216 |
| Cross-shore extent (m) | 134 | 309 | 366 | 650 |
| Average cross-shore cliff extent (m) | 26 ± 4 | 101 ± 42 | 49 ± 40 | 158 ± 56 |
| Average cliff base elevation (m, NAVD88) | 3.3 ± 0.6 | 4.9 ± 1.0 | 3.9 ± 2.3 | 3.6 ± 1.9 |
| Average cliff height (m) | 31 ± 3 | 34 ± 4 | 38 ± 17 | 105 ± 35 |
| Average foreshore slope (°) | 4.4 ± 4.3 | 7.4 ± 7.2 | 7.5 ± 8.5 | 6.8 ± 8.6 |
| Average cliff slope (°) | 50 ± 10 | 30 ± 17 | 38 ± 17 | 36 ± 8.2 |
| Average hinterland slope (°) | 6.9 ± 7.3 | 17 ± 15 | 11 ± 9.2 | 22 ± 10 |
| Morphology | Cliff is the unique slope, beach and hinterland are planar, cliff top has a near-straight shape | Rotational landslides resulting in within-cliff flattening, trees, beach cusps, gullies that intersect the cliff top | Plunging cliff, foreshore rocks, road adjacent to the cliff, sloping hinterland | Steep hinterland, cliff is part of a mountain slope, multiple superposed landslides |

*CliffDelineaTool* and *CliffMetrics* were run on the four test sections (AOIs #5-8) and compared to the manually
mapped cliff lines. For *CliffDelineaTool*, *NVert* was set at 20 because the test sections have the same 1 m DEM resolution as the calibration set. *MaxBaseElev* was set based on average cliff base elevation (Table 4). Remaining parameters values (*BaseSea*, *BaseLand*, *TopSea*, *TopLand*, *PropConvex* and *SmoothWindow*) were selected based on average foreshore, cliff and hinterland slopes (Table 4), and visual assessment on initial model runs (Table 1). RMS between true and modelled cliff base and top locations was used to assess model performance on all transects
(including skipped transects).

## 3 Results

Consistency of the manual, and performance of the automated cliff mapping varied between the four evaluation AOIs. The simple cliff geometry and unambiguous location of the cliff base and top of AOI #5 resulted in low (≤2.7 m) RMS for all mapping methods (Table 5; Figure 7a-b). In general, cliff base locations were more consistent between manual
mappers and *CliffDelineaTool* compared to the cliff top. *CliffMetrics* had high (>>10 m) cliff base RMS for all three complex AOIs, sometimes placing the cliff base on the upper cliff face or hinterland (B in Figure 7d). *CliffDelineaTool* (RMS = 1.1-8.2 m) and manual mapping (RMS = 0.7-11 m) gave comparable cliff base detection results. For AOI #6 *CliffMetrics* sometimes placed cliff base at beach cusps (C in Figure 7d).

For complex AOIs #6-8, manual and automated cliff top positions were varied (RMS = 2.9-214 m). In AOI #6 one
mapper (#2) selected the head scarp of an interior landslide as the cliff top about 100 m from the cliff top selected by all other mappers (A in Figure 7c). However, both models also picked sections of the landslide head scarp (D in Figure 7d). In AOI #7 two mappers interpreted an elevated section between two separate landslide scars as cliff top (E in Figure 7e), differing from most other mappers and both models that opted for a simpler cliff top shape (Figure 7e-f). The cliff top position was the most diverse for AOI #8, with mappers' interpretation ranging from the top of the coastal
mountain slope (F in Figure 7g) to the mid-slope (G in Figure 7g), and tops of interior cliff face landslide scars (H in



Figure 7g). *CliffDelineaTool* gave inconsistent results, while *CliffMetrics* placed the cliff top at the top of the mountain

slope (Figure 7h).

**Table 5. RMS between the true (median of manually mapped), individual manual mapping and modelled cliff base and top positions.**

|  | AOI #5 | | AOI #6 | | AOI #7 | | AOI #8 | |
|---|---|---|---|---|---|---|---|---|
|  | base | top | base | top | base | top | base | top |
| Mapper #1 | 0.9 | 1.1 | 2.5 | 8.1 | 11 | 2.9 | 3.6 | 174 |
| Mapper #2 | 1.0 | 1.5 | 4.3 | 28 | 3.6 | 6.8 | 3.0 | 79 |
| Mapper #3 | 1.1 | 1.3 | 1.5 | 10 | 4.8 | 18 | 4.2 | 166 |
| Mapper #4 | 0.8 | 1.7 | 1.2 | 7.5 | 1.8 | 5.3 | 2.9 | 116 |
| Mapper #5 | 0.8 | 1.2 | 1.3 | 10 | 2.7 | 13 | 2.7 | 25 |
| Mapper #6 | 1.4 | 1.3 | 4.3 | 9.0 | 6.6 | 21 | 5.8 | 111 |
| Mapper #7 | 1.1 | 1.7 | 1.2 | 16 | 5.6 | 4.2 | 5.1 | 61 |
| Mapper #8 | 1.0 | 1.5 | 1.6 | 11 | 3.5 | 4.1 | 3.7 | 169 |
| Mapper #9 | 0.9 | 2.3 | 1.3 | 11 | 3.4 | 3.9 | 2.3 | 49 |
| Mapper #10 | 0.7 | 1.9 | 2.7 | 13 | 5.2 | 5.4 | 2.5 | 17 |
| Average of mappers | 1.0 | 1.5 | 2.2 | 12 | 4.8 | 8.5 | 3.6 | 97 |
| *CliffMetrics* | 2.1 | 2.7 | 82 | 32 | 82 | 28 | 63 | 215 |
| *CliffDelineaTool* | 1.1 | 1.5 | 3.8 | 25 | 8.2 | 8.3 | 4.8 | 99 |





**Figure 7.** (a, c, e, g) Individual manually mapped and (b, d, f, h) median of manual mapping and modelled cliff base and top positions for AOIs #5-8. Locations A: pre-slide cliff top position interpreted as cliff top by mapper #2; B: cliff base placed in the hinterland by *CliffMetrics*; C: beach cusps identified as cliff base by *CliffMetrics*; D: interior cliff face location selected as cliff top by *CliffDelineaTool* and *CliffMetrics*; E: elevated section between two landslide scars interpreted as cliff top by two mappers; F: top of the mountain slope interpreted as the cliff top; G: cliff top placed in the mid-slope; H: cliff top placed at the lower mountain slope.



## 4 Discussion

Acceptable model results depend on the purpose of cliff delineation. At the scale and resolution considered here, the model generally provided comparable results to manual mapping for diverse cliff morphology. Model performance

generally correlates with the amount of inconsistency between manual mappers related to the cliff complexity. For example, in complex AOI #8 mean manual mapper and *CliffDelineaTool* RMS were both high (97 and 99 m, respectively). Conversely, for simple AOI #5, manual mapper and *CliffDelineaTool* RMS were both low (1.5 m both). The present model does not resolve the situations where a transect crosses the cliff top multiple times, and typically places the cliff top at the seaward-most crossing point (Figure 6b). Other model shortcomings include occasional

treetop selection for cliff top positions (Figure 6c-d). Given these issues and high RMS in complex cliff sections we suggest that model outputs should be visually controlled (similar to Payo et al., 2018), and not used where the cliff top is very ambiguous such as AOI #8 with complex tall mountain slopes.

The input parameters have a varied impact on model performance (Figure 6). *MaxBaseElev* is easily selected using a slightly conservative cliff base elevation estimated from the general site settings. The optimal *NVert* parameter

depends on DEM resolution and cross-shore cliff extent. Calibration showed that in general the greater the *NVert* value the narrower the distribution of the four threshold slopes (*BaseSea*, *BaseLand*, *TopSea* and *TopLand*) and the higher the number of outliers. That relationship holds until *NVert* value (30 for AOI #1, 40 for #3 and 70 for #4, Figure 4) over which the $q_1$ - $q_3$ box becomes very wide or the distribution is dominated by outliers. Figure 8 shows the cliff top optimized for *NVert* values of 10, 20, 30, 40 and 50 for AOI #4. It suggests that during stage 2, low *NVert* value

can cause incorrect cliff top section caused by minor protruding features such as vegetation (A in Figure 8), while higher value can inhibit correct stage 2 cliff top replacement (B in Figure 8). *TopLand* of 10-20° is generally appropriate when hinterland is planar or gently sloping but should be increased for cliffs with steeper inland areas. *BaseSea* and *BaseLand* are important for proper cliff base placement for coastal sections with back beach cusps and landslide deposits. Model results are sensitive to *TopSea*, *PropConvex* and *SmoothWindow*, and testing various values

on short sections of the study area with visual inspection can help identify optimal values (Figure 6).



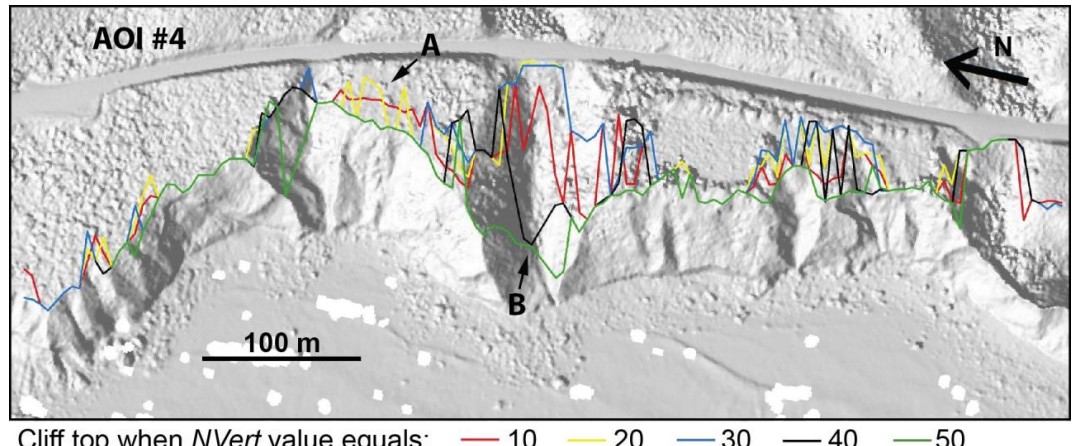

**Figure 8. Location of the stage 2 cliff top after optimizing all parameters for *NVert* ranging 10-50. Location A: vegetation interpreted as cliff top for model runs with low *NVert* values. Location B: high *NVert* values can inhibit correct stage 2 cliff top replacement to locations further landward.**

Two previous studies have successfully used parts of *CliffDelineaTool*. Swirad and Young (2021) used a modified version of *CliffDelineaTool* to automate mapping of cliff base and top positions along the California coast (1646 km). The modified version did not include stage 2 (shifting cliff top landwards for within-cliff flattening areas) and stage 3 (removal of outliers) but did include a Laplacian topographic filter (Richter et al., 2013). The automated results were visually inspected and some (10% of the cliff base and 29% of the cliff top positions) required manual modification to correct positions. Young et al. (2021) used the present *CliffDelineaTool* model to identify the cliff base in 155 0.25 m resolution DEMs along a 2.5 km coastal section at 1 m alongshore transect spacing. Quality control showed that cliff base misplacement was negligible, while the total processing time was ~30 min (Young et al., 2021). These studies demonstrate the tool's applicability for both large space and time datasets, and over a range of DEM resolutions.

## 5 Conclusions

Building on previous studies, we developed a new algorithm (*CliffDelineaTool*) to delineate coastal cliffs from DEMs. The model identifies cliff base and top positions along cross-shore transects using elevation and slope characteristics. It considers complex cliff morphology and removes alongshore cliff top outliers. *CliffDelineaTool* provides results comparable to manual mapping and outperforms existing models for the complex cross-profiles analyzed. The automated results have known errors and should be inspected visually. The method has been applied successfully on two large datasets (Swirad and Young, 2021; Young et al., 2021), greatly reducing processing time. With calibration and quality control *CliffDelineaTool* can be used on a wide variety of coastal setting facilitating a range of scientific and managerial applications but has limited application where the cliff top is ambiguous.

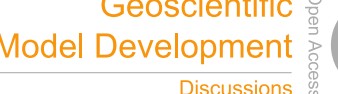

## Code availability

*CliffDelineaTool* was coded in MATLAB R2019a. The source code, and the calibration and validation datasets explored in this paper (DEMs, model input, true and modelled cliff base and top positions) are available at https://zenodo.org/record/5510191 (Swirad, 2021). See https://github.com/zswirad/CliffDelineaTool (last access: 2021-09-15) for the latest version of the source code and user instructions.

## Acknowledgements

The project was funded by the California Ocean Protection Council (C0303100) administered by University of Southern California Sea Grant. Additional support was provided by the California Department of Parks and Recreation, Natural Resources Division Oceanography Program (C1670004 and C19E0049). We used freely available topographic data by NOAA Office for Coastal Management (coast.noaa.gov). We thank researchers at the Center for Coastal Studies (Scripps Institution of Oceanography) for helping validate the algorithm and Gregor Lützenburg (University of Copenhagen) for providing feedback on its early version.

*Author contribution:* Conceptualization, methodology, formal analysis: ZMS. Funding acquisition: APY. Writing, visualization: ZMS & APY.

*Competing interests:* The authors declare that they have no conflict of interest.

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
