# Peer review of "CliffDelineaTool v1.2.0: an algorithm for identifying coastal cliff base and top positions"

_Geoscientific Model Development, 2021_

## Author Response (AR1)

Dear Editor and Referees,

Thank you for your consideration of this manuscript and insightful reviews that we believe significantly improved the manuscript. Below we address each comment and describe the changes made to the manuscript. Our responses are in blue, and the line numbers refer to the clean revised manuscript.

Sincerely,
Zuzanna Swirad & Adam Young

**RC1: ANONYMOUS REFEREE #1**

Thank you so much for the opportunity to review the manuscript entitled CliffDelineaTool v1.1.0: an algorithm for identifying coastal cliff base and top positions by Zuzanna M. Swirad and Adam P. Young. The manuscript is well written and organized.

The manuscript presents MATLAB-based algorithm that identifies cliff base and top positions on complex cliffs using cross-shore transects extracted from digital elevation models named CliffDelineaTool (v1.1.0). The advantages of having an automated method to extract cliff metrics is obvious and probably no more evident than when change analysis or erosion rates are to be computed from different lidar surveys.

The authors are basing their work on work previously published by Palaseanu-Lovejoy et al. (2016) and Payo et al. (2918) adding to the narrative few refinements in order to process complex cliff profiles and increase accuracy. While the previous authors algorithms were published in completely open-source software this work is presented in MATLAB, a proprietary software that necessitate a license. This unfortunately is a drawback of the present manuscript.

Both Palaseanu-Lovejoy et al. (2016) and Payo et al. (2018) present techniques to establish transects approximately perpendicular to the cliff face as well as ways to decide the length of these transects. Actually, in this, the two above mentioned papers differ the most, besides the programing language used. The present manuscript of Swirad and Young is not explicit on how the transects are generated although some indication on how their length is established is presented by discussing the user defined parameter NVert. Palaseanu-Lovejoy et al. (2016) has two main user defined parameters, the length of shoreline that is approximately parallel with the cliffs that have a comparable elevation range and one buffer distance for each individual shoreline segment in the effort to compensate against widely variable cliff elevation ranges to increase accuracy to top and toe delineation and decide on transect lengths. Payo et al. (2018) chose to forgo the necessity of buffers and use a constant transect length to speed up the time of processing considerably with some accuracy loss.

While both Palaseanu-Lovejoy et al. (2016) and Payo et al. (2018) have relatively few user-defined parameters, Swirad and Young method has 8 user-defined parameters that need to be established in accordance with different morphologies.

Swirad and Young method concentrates on evaluating other top candidates on very complex cliff profiles, and after using an outlier filter (Tukey ,1977) method to eliminate them. Although previous work does not necessarily look in details to other top candidates on complex profile, the work of Palaseanu-Lovejoy et al. (2016) does mention both secondary tops or toes, as well as using the same method of eliminating outliers from top / toe results as in this manuscript. The same work from 2016 has checks within the presented algorithm that does not permit on complex profiles to have toe positions that are either at higher elevation than the top of the cliff or more in-land on the transect than the position of the top of the cliff.

This internal check is missing, it seems, from the CliffMetrics tool of Payo et al (2018) that was run on the present authors examples as demonstrated in manuscript Figure 7.

Very recently Plalaseanu-Lovejoy published the R code that follows the 2016 publication:  Palaseanu-Lovejoy, M., 2021, iBluff - Geomorphic analysis of coastal bluffs / cliffs. https://doi.org/10.5066/P9HJ7QHD This code has functions to eliminate outliers using Tukey (1977) method, generating secondary points of inflection (secondary top / secondary toe) on the cliff face, define all "positive tops" and "negative toes" on a transect (in the code called dunes crests and troughs), and get the relative convexity and concavity of the cliff / bluff profile.

In conclusion this work is interesting and tries to solve the problem that the method is relatively sensitive to cliff / bluff morphology in extreme complex cliff profiles. Also this work demonstrates that the definition of what constitute both cliff top and toe can differ from one analyst to another, from one location to another and from one geomorphology to another.

Thank you for the thorough and constructive comments on our study considering previous models. We address all issues described above in reference to the specific comments listed below.

**Specific comments**

Ln 49: What are the advantages to develop this new tool in MATLAB vs. an open-source software, when MATLAB is a proprietary software that necessitate a license? Can the algorithm be transferred in Python, R, or C++?

We originally developed the tool in MATLAB while working on a California coastal cliff erosion assessment (Swirad and Young, 2021). We agree that using open-source software has many advantages and have coded the tool in Python as suggested. The code is now available on GitHub (https://github.com/zswirad/CliffDelineaTool).

In the manuscript we updated the abstract (lines 8-9):
"Here we describe *CliffDelineaTool* (v1.2.0), a MATLAB/Python-based algorithm…"

We also modified the 'Code availability' section to (lines 271-273):
"See https://github.com/zswirad/CliffDelineaTool (last access: 2021-11-24) for the latest version of the source code, Python version of the tool (*CliffDelineaToolPy*, v1.0.0) and user instructions."

Ln 75: In Figure 2, a, b, and c the vertical axis is approx. 4 to 5 times exaggerated in comparison with the horizontal axis. This makes the slope to appear steeper than it is. There is approx. 20 m elevation increase over ~ 100 m cross-shore distance, so the slope cannot be more than 10 - 12 degrees. In some instances that could be a sand dune in front of a cliff, like on the east shore of Lake Michigan. The authors are right that sometimes the definition of a cliff / bluff can change with location, so maybe a photograph from the location where this transect is, might help to prove that the whole profile is of a cliff, and not a cliff and a dune in front of it.

This is a very good comment. As suggested, we added airborne photographs of the eight AOIs in Figure 3 and in Figure 2 caption refer to the map and the photograph (lines 77-78):

"see Figure 3c for topography and airborne photograph"

The vertical exaggeration in Figure 2a-c was used for better visualization. We have added a note in the figure caption (line 83):

"Note the vertical exaggeration in a-c and horizontal exaggeration in d-f."

Ln 104: What are the parameters of the moving window? A 20 m vs. 50 m moving window will give different results, theoretically.

Correct. We discuss the moving window parameters in section 2.2.1 in lines 153-155:

"Optimal *SmoothWindow* was selected by comparing moving window ranging from 1 to 20 alongshore transects for each AOI to minimize RMS and number of skipped transects (Figure 5e-h; Table 1)."

Calibration of the *SmoothWindow* parameter is discussed in lines 241-244:

"*SmoothWindow* parameter depends on alongshore complexity of the coast, bay/headland sequence spacing, and their relation to transect spacing. Model results are sensitive to *TopSea*, *PropConvex* and *SmoothWindow,* and testing various values on short sections of the study area with visual inspection can help identify optimal values (Figure 6)."

Ln 106: What means in this instance "re-examined"? Is the algorithm to be run again to get an alternative top? It is not clear from this context.

Yes, we now clarify this in lines 110-112:

"Transects with standardized residuals (residual divided by residual standard deviation) >2 are flagged as outliers (Figure 2e) and re-examined to identify new potential cliff top locations."

Ln 111: Are these coastal cliff sections relevant for other locations such as northern shore of Alaska or the bluffs around Lake Michigan? Or for each location with different geological context the calibration exercise needs to be repeated?

Good comment. We clarify this in line 230:

"For best performance, *CliffDelineaTool* should be calibrated to the user study section."

Ln 115: It seems that the transects are approx. 5 m apart for each AOI. The DEM is at 1 m resolution so theoretically you can have transects 1 m apart without repeating the cliff data profile. Was there any reason to decide to have transects 5 m apart and not 3 m or 1 m or 10 m apart? How where the transects generated? In Palaseanu-Lovejoy et al (2016) and Payo et al (2018) the generation of transects is explained.

Thank you for the comment. We have modified the manuscript for clarification. Generation of transects is not part of the tool. The transects are a tool input. Lines 62-63 read:

"A single input file includes multiple ordered transects representing an entire coastal section."

We explain why we selected 5 m alongshore spacing in lines 126-127:

"Parallel calibration transects generated with ArcGIS tools were spaced 5 m alongshore to capture meso-scale details of alongshore cliff geometry and sampled at 1 m cross-shore resolution."

We discuss generation of transects in lines 217-221:

"Unlike the models of Palaseanu-Lovejoy et al. (2016) and Payo et al. (2018), *CliffDelineaTool* does not generate transects. For this study, simple cross shore transects with 5 m alongshore spacing were defined manually for each AOI. The relatively short AOI alongshore sections permitted use of parallel transects (Swirad, 2021). However, for longer, more complex cliff sections varying transect orientation will improve results and could be generated with *CliffMetrics* (Payo et al., 2018)."

Ln 120: This is a very informative figure and maybe it should be placed somewhere more towards the beginning of this section. Also for each profile graph you should label on the y axis the vertical exaggeration, since this impacts how we perceive the cliff slope.

Thank you. We have labeled the vertical exaggeration as suggested (Figure 3). Figure 3 is located after the first paragraph of the section and just after the first time it is referenced according to standard formatting.

Ln 124: 3, 4 and 9 m are all for AOI #2 or AOI #2 to #4?

As suggested, we clarified lines 131-133:

'Maximum cliff base elevation (*MaxBaseElev*) was subjectively set to 5 m (NAVD88) for AOIs #1, #3, and #4, and 9 m (NAVD88) for AOI #2 based on DEM inspection (Figure 3, Table 2).'

Ln 126: What this means? Do you mean eliminating outliers?

We are unsure which statement the reviewer refers to. We assume it is '[…] while not skipping too many transects.' Transects are skipped if no point fulfils the potential cliff base or top criteria. This is explained in lines 89-90 and 96-97 that read:

"If no points fulfill the [cliff base] criteria, the transect is skipped at this stage."

"If no points along the transect fulfill the [cliff top] criteria, the transect is skipped and has no explicitly modelled cliff top."

Ln 129: This is the basic understanding of the box-and-whisker plots and outlier interpretation from Tukey, John W (1977). Exploratory Data Analysis. Addison-Wesley. Please add citation.

Citation added as suggested (line 139).

Ln 132-133: This sentence implies that the black whisker ends are the outliers, when the outliers are the values beyond the black whisker ends that are represented by red crosses. Please re-phrase.

As suggested, we re-phrased lines 140-142:

"Threshold slope angles were picked as minimum or maximum values excluding outliers ($q_3 + 1.5 \times (q_3 - q_1)$ or $q_1 - 1.5 \times (q_3 - q_1)$; black whisker ends in Figure 4; Table 1)."

Pg. 14, Table 5: These results are very interesting. For CliffMetrics did you run the Payo et al (2018) code? That code does not have a routine/function to eliminate outliers, while Palaseanu-Lovejoy et al (2016) has a function to eliminate outliers from both top and toe positions using Tukey (1977) method, same method that is used in this manuscript as well. Very recently Plalaseanu-Lovejoy published the R code that follows the 2016 publication: Palaseanu-Lovejoy, M., 2021, iBluff - Geomorphic analysis of coastal bluffs / cliffs. https://doi.org/10.5066/P9HJ7QHD

This R code has a function to eliminate outliers.

Also CliffMetrics from Payo et al (2018) uses a constant length transect for an area, does not matter how big or small that cliff elevation range is. If there is a cliff at a base of a hill or mountain and that transect is so long that reaches the top of the hill or mountain then the CliffMetrics algorithm could select that top instead of the cliff top. This was a decision made by the authors to speed up the process of deciding how long those transects should be from the original Palaseanu-Lovejoy et al (2016) method where the length of the transects were a user defined parameter for sections of relatively comparable cliff ranges. The algorithm presented in this manuscript eliminates outliers, so the comparison is between 2 methods that use the same logic to select an initial top / toe but one eliminates outliers and one does not. So it is logic to get better results for the method that eliminates outliers.

What happens if you compare the CliffMetrics method with stage 1 of this manuscript method? Or compare stage 2 or 3 results from this manuscript with Palaseanu-Lovejoy et al (2016) and Palaseanu-Lovejoy (2021) code that eliminates outliers.

In conclusion I do agree, you will always get better results when outliers are eliminated.

Thank you for this comment. We have now clarified which models we compare *CliffDelineaTool* to in lines 128-130:

"The *CliffDelineaTool* results were compared with the distance-to-trendline method (Palaseanu-Lovejoy et al., 2016) and *CliffMetrics* (SAGA GIS version; Payo, 2020) using input parameters (seaward transect end points, transect length, and no transect smoothing) to match the same cross shore transects used for *CliffDelineaTool* and the default vertical tolerance of 0.5."

Furthermore, we discuss the differences in the basic 'distance-to-trendline' method and the models of Palaseanu-Lovejoy (2021) and Payo et al. (2018) in lines 222-229:

"We compared *CliffDelineaTool* to the distance-to-trendline method which forms basis of *iBluff* (Palaseanu-Lovejoy, 2021) and *CliffMetrics* (Payo et al., 2018). However, *iBluff* and *CliffMetrics* both include additional steps to improve results and correct erroneous cliff base and top positions. *iBluff* uses manual transect shortening during pre-processing, and outlier removal using smoothing window, similar to *CliffDelineaTool. CliffMetrics* uses manual quality control and iterative parameter selection (Payo et al., 2018). The *CliffMetrics* results presented here used default parameters and predefined transects to provide a direct comparison to *CliffDelineaTool.* However, one of the strengths of *CliffMetrics* includes the ability to quickly iterate parameter set up. Therefore, the results could be improved using iterative parameter selection and varying transect length and orientation."

We have added the reference to Palaseanu-Lovejoy (2021) in lines 44-45:

"Palaseanu-Lovejoy (2021) updated the model (*iBluff*, coded in R) to include automatic outlier removal using a moving window (Tukey, 1977)."

Ln 190: Figure 7. A very nice and informative figure. The length of the transects were the same for both the CliffMetrics and the CliffDelineaTool? For example, in figure e and f there is a cliff face that has a vertical range about 4 to 5 times bigger that the cliff faces on both left and right of it. It seems that a very long transect was used with the CliffMetrics algorithm from Payo et al (2018). Palaseanu-Lovejoy et al (2016) algorithm would have used different transect lengths and would check that the toe elevation and position is not higher and more in-land than the top elevation and position on the same transect, and if it is a new one is either selected or the toe is rejected completely for that transect. From the figure f it seems

that the CliffMetrics selected a toe that is more in-land than the top of the cliff in that instance (same for figure d, the B situation).

Thank you for this comment. We have now clarified lines 223-226 to state:

"However, *iBluff* and *CliffMetrics* both include additional steps to improve results and correct erroneous cliff base and top positions. *iBluff* uses manual transect shortening during pre-processing, and outlier removal using smoothing window, similar to *CliffDelineaTool*. *CliffMetrics* uses manual quality control and iterative parameter selection (Payo et al., 2018)."

Ln 204 – 205: I don't consider this a shortcoming of the method. The method requires a bare-earth DEM, if some vegetation is still present in the DEM it is obvious that the top of a tree close to the top of the cliff might be identified as the cliff top. This is a shortcoming of the DEM input data and not of the method per se.

We have added the note in lines 213-214:

"Other model application issues include occasional treetop selection for cliff top positions (Figure 6c-d) when using non-bare-earth DEMs."
* * *
**RC2: DOMINIK PAPROTNY**

"CliffDelineaTool v1.1.0: an algorithm for identifying coastal cliff base and top positions" is a manuscript that presents the validation of a Matlab-based tool for coastal applications. As the other reviewer discussed the manuscript and other possible methods quite extensively, and further I find the text and figures adequate apart from the issues already raised, I decided to focus on the code.
The paper extensively discusses the accuracy and calibration issues of the method – especially interesting is the comparison with results obtained by different human mappers. I tested the tool using a 1-meter lidar DEM taken in 2011 by the national surveying agency around the BiaÅ,  a Góra cliffs near MiÄ™dzyzdroje, north-western Poland (fragment of a DEM used in: Paprotny and Terefenko, 2017, Natural Hazards, DOI 10.1007/s11069-016-2619-z). The area is rather tricky due to a very narrow beach, a lot of landslides and anthropogenic influence. I generated 55 cross-shore transects using ArcMap as per the instructions on GitHub. The script run without problems on those transects. After visualizing the output in ArcMap (see attachment), even though the parameters were kept at default, the cliff base and top were identified very well most of the time.

We are very grateful to the reviewer for taking time to run the code and evaluate the results on an independent data set. We are encouraged that the model gave good results using default settings without local calibration despite the different morphology.

I have some minor suggestions about the code:

As suggested, we have modified the code following reviewer's comments. The code has a new release (v1.2.0) and manuscript title has been updated.

Making the code a function script would make it slightly faster to embed the tool in the workflow by other researchers.

As suggested, we converted the code to a function.

In line 37, use "display" function

Modified as suggested (code line 39).

In line 39, showing "table" is not needed as it will be always empty here;

Removed as suggested.

It would be good to add a header to the output file, so it is immediately known what the columns contain; also, it would make it easier to load in software other than ArcMap, e.g. QGIS.

As suggested, we have added the option to save the files with column names (code lines 358-369 for cliff base and 392-403 for cliff top).

Further, it would be much more convenient if the script could read XY coordinates of those (if included in the attribute table – could be an optional parameter) and save the output with those, so it can be directly viewed in GIS without the need to merge it with the shapefile every time.

We have modified the code so that transect point attributes such as X, Y coordinates included in the original input file can be maintained through the tool and saved in output files (code lines 342-353 and 376-387).

As a future development, that I think is beyond the scope of this paper, I would suggest to:
Create a Python script that would contain the pre-processing routine of ArcMap. All except defining the seaward and landward limits as polylines can be rendered in Python.
Further, use geopandas in such a script to export the attribute table and then post-process the Matlab output directly into a shapefile.

Thank you for the suggestions. The code is now available in Python. Future versions and codes to automate pre- and post-processing will be available on GitHub.

I'm looking forward to the authors' revision of the paper and code.
* * *
**CC1: ANDRES PAYO**

Thanks for the opportunity to comment on your manuscript. I am Andres Payo, lead developer of CliffMetrics and would very much like to see the comparison with your CliffDeliea Tool but I have one major concern with the current version of the manuscript which is regarding the lack of information about which version, and software and set up you have used to create the CliffMetrics outcomes. In addition to the code added to the GMD Payo et al. (2018) manuscript, CliffMetric is also available via SAGA tools (URL = http://www.saga-gis.org/saga_tool_doc/7.9.0/ta_cliffmetrics_0.html). Which version have you used for this work is unclear. Most importantly, which set up have you used is also unclear. I would appreciate if you include the input values as shown in Table 6 Payo et al. 2018 or SAGA input table. Some of the jagginess that you seem to obtain with CliffMetrics (your Figure 7) are could easlily be avoided by iterating the CliffMetric set up parameters. CliffMetric runs fast to facilitate the iterative delineation of the cliff top and toe. Your own method has this iteration embeded in the methodology. As the manuscript stands now, I can not tell if your method is performing better than CliffMetric or you are just miss-using CliffMetrics by using the wrong iterative set-up.

Thank you for your insightful and helpful comments. We initially used the 'distance-to-trendline' method that is the basis of Palaseanu-Lovejoy et al. (2016) and Payo et al. (2018). We have now added *CliffMetrics* using default parameters for a more direct comparison. We now clarify this in lines 128-130:

"The *CliffDelineaTool* results were compared with the distance-to-trendline method (Palaseanu-Lovejoy et al., 2016) and *CliffMetrics* (SAGA GIS version; Payo, 2020) using input parameters (seaward transect end points, transect length, and no transect smoothing) to match the same cross shore transects used for *CliffDelineaTool* and the default vertical tolerance of 0.5."

We added *CliffMetrics* RMS (default settings) to Tables 3 and 5, as well as plotted it in Figure 7.

We further discuss how using *iBluff* and *CliffMetrics* would impact the distance-to-trendline method in lines 222-229:

"We compared *CliffDelineaTool* to the distance-to-trendline method which forms basis of *iBluff* (Palaseanu-Lovejoy, 2021) and *CliffMetrics* (Payo et al., 2018). However, *iBluff* and *CliffMetrics* both include additional steps to improve results and correct erroneous cliff base and top positions. *iBluff* uses manual transect shortening during pre-processing, and outlier removal using smoothing window, similar to *CliffDelineaTool*. *CliffMetrics* uses manual quality control and iterative parameter selection (Payo et al., 2018). The *CliffMetrics* results presented here used default parameters and predefined transects to provide a direct comparison to *CliffDelineaTool*. However, one of the strengths of *CliffMetrics* includes the ability to quickly iterate parameter set up. Therefore, the results could be improved using iterative parameter selection and varying transect length and orientation."

Minor concerns:
In Page 2 Line 45, the following sentence is not true "They used a constant transect length with decrease in model performance, but considerable time gain (Payo et al., 2018)." We did not found a decrease in model performance relative to PL2016 model and we explicitiky indicate that "By avoiding the need for fine-tuning the profile length, the proposed method speeds up the delineation process but does not eliminate the need for the screening of the model outputs." Please clarify what do you mean regarding decrease in model performance.

As suggested, we modified lines 47-48 to:

"Payo et al. (2018) used a constant transect length to reduce pre-processing time."

---

## Referee Report (RR1)

Thank you for giving me the opportunity to review the revised manuscript CliffDelineaTool v1.2.0: an algorithm for identifying coastal cliff base and top positions by Zuzanna M. Swirad and Adam P. Young.

The revised manuscript answered sufficiently the reviewers' comments and suggestions. I am definitely impressed with submitting a Python code that replicates the MATLAB code that was the initial basis for this manuscript since MATLAB is a licensed software and Python is open-source code and hence can accommodate a wider user group.

Figure 3 is now clearer showing not only the vertical exaggeration on profiles, but also the images of selected locations to get a better idea of the morphology, especially for scientists not familiar with those particular locations of the California Coast.

Table 3 is extremely informative for the three-method comparison. It is interesting to see that for relatively simple profiles for AOI 1 and 2 at least for the top of the cliffs the results are very comparable, while for more complex profiles usually cliff top selection performs better than cliff toe selection, and at least at the stage 2 level of CliffDelineaTool. I would expect improved results in the other methods if the additional code is run to eliminate outliers and refine their code parameters, while this table shows results when running the code with default parameters.

Lines 230 to 245 are a nice comparison synopsis of the three considered methods, especially since all three basically use the initial methodology described by Palaseanu-Lovejoy et al (2016) to select the base and top of a cliff on transects but differ in how each method threats generating the transects, outliers and complex cliff profiles. The authors of the CluiffDelineaTool made the effort to incorporate in the main function outlier elimination and selection rules to choose either base or top of the cliff when multiple candidates are available, while the methods presented by Palaseanu-Lovejoy (2021) and Payo (2020) leave the decision to the user if they will run the extended code to improve results. Ultimately all three methods need a final visual inspection to ensure the quality of the results.

In conclusion the revised manuscript is improved in clarity and presentation and I am suggesting to accept it for publication.

---

## Author Response (AR2)

Dear Editor,

The slight difference in model results when using MATLAB vs Python versions of the code are related to the stage 3 cliff top identification, namely alongshore outlier removal. We are now stating it in Code Availability as you suggested (lines 273-274):

"Cliff top positions identified with MATLAB and Python versions of the code may vary slightly due to the program-specific outlier removal functions."

Kind regards,

Zuzanna Swirad & Adam Young